# Trends in Hospital Admissions and Death Causes in Patients with Systemic Lupus Erythematosus: Spanish National Registry

**DOI:** 10.3390/jcm10245749

**Published:** 2021-12-08

**Authors:** Víctor Moreno-Torres, Carlos Tarín, Guillermo Ruiz-Irastorza, Raquel Castejón, Ángela Gutiérrez-Rojas, Ana Royuela, Pedro Durán-del Campo, Susana Mellor-Pita, Pablo Tutor, Silvia Rosado, Enrique Sánchez, María Martínez-Urbistondo, Carmen de Mendoza, Miguel Yebra, Juan-Antonio Vargas

**Affiliations:** 1Internal Medicine Department, Health Research Institute Puerta de Hierro-Segovia de Arana (IDIPHIM), Hospital Universitario Puerta de Hierro Majadahonda, 28222 Madrid, Spain; rcastejon.hpth@salud.madrid.org (R.C.); Angelagutierrezrojas@gmail.com (Á.G.-R.); Pedrodurandc@hotmail.com (P.D.-d.C.); Susanamellor@hotmail.com (S.M.-P.); pablo.tutor@hotmail.com (P.T.); Silvia.rosado@salud.madrid.org (S.R.); sanchezchica@gmail.com (E.S.); mmurbistondo@gmail.com (M.M.-U.); cmendoza.cdm@gmail.com (C.d.M.); myebrab@hotmail.com (M.Y.); juanantonio.vargas@salud.madrid.org (J.-A.V.); 2Basic Medical Sciences, Faculty of Medicine, Universidad CEU San Pablo, 28003 Madrid, Spain; carlos.tarincerezo@ceu.es; 3Autoimmune Diseases Research Unit, BioCruces Bizkaia Health Research Institute, UPV/EHU, 48903 Bizkaia, Spain; r.irastorza@outlook.es; 4Clinical Biostatistics Unit, Health Research Institute Puerta de Hierro-Segovia de Arana, CIBERESP, 28222 Madrid, Spain; aroyuela@idiphim.org

**Keywords:** systemic lupus erythematosus, cardiovascular disease, infections, neoplasm, mortality, hospital admissions

## Abstract

Background: the admission and death causes of SLE patients might have changed over the last years. Methods: Analysis of the Spanish National Hospital Discharge database. All individuals admitted with SLE, according to ICD-9, were selected. The following five admission categories were considered: SLE, cardiovascular disease (CVD), neoplasm, infection, and venous-thromboembolic disease (VTED), along four periods of time (1997–2000, 2001–2005, 2006–2010, and 2011–2015). Results: The admissions (99,859) from 43.432 patients with SLE were included. The absolute number of admissions increased from 15,807 in 1997–2000 to 31,977 in 2011–2015. SLE decreased as a cause of admission (from 47.1% to 20.8%, *p* < 0.001), while other categories increased over the time, as follows: 5% to 8.6% for CVD, 8.2% to 13% for infection, and 1.4% to 5.5% for neoplasm (*p* < 0.001 for all). The admission mortality rate rose from 2.22% to 3.06% (*p* < 0.001) and the causes of death evolved in parallel with the admission categories. A significant trend to older age was observed over time in the overall population and deceased patients (*p* < 0.001). Conclusions: Better control of SLE over the past two decades has led to a decrease in early admissions, and disease chronification. As a counterpart, CVD, infections, and neoplasm have become the main causes of admissions and mortality.

## 1. Introduction

Systemic lupus erythematosus (SLE) is an extraordinarily complex disease with a wide variety of clinical features and phenotypes [1]. SLE is a chronic disease mainly affecting young women who suffer from chronic inflammation, disease flares, cumulative drug toxicity, and frequent hospital admissions [2,3,4,5]. Since the last century, huge advances have been made in the diagnosis and management of lupus [6]. In parallel with the development of more efficient and less harmful treatments, great concern has emerged related to the long-term complications of SLE, such as the high prevalence of cardiovascular disease, the risk of infections, and the impact on quality of life [7]. Therefore, important changes in the relative weights of the different causes of hospital admissions of lupus patients are likely to have taken place over the last years.

In light of the foregoing, our objective was to analyze the trends in the hospital admissions and mortality of Spanish SLE patients over the last two decades.

## 2. Materials and Methods

We performed a registry study with a case series design, where the hospital admissions and death causes in SLE patients were the main outcomes. We analyzed the data extracted from the Spanish Hospital Discharge Database (SNHDD), a national registry belonging to the Spanish Ministry of Health. The SNHDD includes demographic and epidemiological data and up to 20 discharge diagnoses carried out during the admission of patients, as coded by the International Classification of Diseases (ICD-9), between 1 January 1997 and 31 December 2015. The study was approved by local research ethics committees (PI_80-21) in accordance with the Declaration of Helsinki. Before making it available to researchers, the database was anonymized and all potential patient identifiers were eliminated.

We selected hospital admissions of patients with a diagnosis within the CIE-9-ES code 710.0 (systemic lupus erythematosus), regardless of its position within the diagnoses coding list. According to the database, the main diagnosis during admission or at discharge was the cause of the admission and/or death. Thus, all the main diagnoses were decoded and classified into the following 5 main groups: active SLE; cardiovascular disease (CVD, comprising coronary disease, cerebrovascular disease, heart failure, arterial thromboembolism, hypertensive kidney disease, arteriosclerosis or peripheral arterial disease); neoplasm (including solid organ, hematological, benign or unknown origin neoplasm); infection (classified according to the foci); venous thromboembolic disease (VTED). Only these admission causes were evaluated. In order to analyze the epidemiological trends over time, we grouped admissions within the following four periods: 1997–2000, 2001–2005, 2006–2010 and 2011–2015.

### Statistical Analysis

Quantitative variables were expressed as mean and standard deviation or as median plus interquartile range (IQR), as appropriate; qualitative variables were expressed as percentages. Numerical variables were compared using the t-test or Mann–Whitney’s U test, and these tests were also used for the analysis of the average age and stay of the patients in the different periods. Normality (Shapiro’s) and homoscedasticity (Levene’s) tests were performed to characterize the populations and therefore Kruskal–Wallis test with post hoc FDR correction was carried out. Categorical variables were compared using the chi-square test. For all the analyses, a significance level of 0.05 was set. Analysis was performed using R and R Studio 1.3.1093.

## 3. Results

### 3.1. Population Characteristics

From a total of 66,462,136 nationwide hospital admissions recorded during the study period, 99,859 involved 43,432 patients with a diagnosis of SLE, according to ICD-9, who were, thus, the object of the analysis. The main descriptive variables are shown in Table 1. The mean age was 46.5 years, 83.3% of the patients were female, the average stay lasted 9.1 days, and the readmission rate was 17.6%. Overall, 2786 individuals (6.41%) died, with an overall admission mortality of 2.79%. During the study period, the mean age upon admission rose from 41.1 years in 1997–2000 to 51 years in 2011–2015, as did the mortality rate (from 4.22% to 5.67%) and the mortality during admission (from 2.22% to 3.06%) (*p* < 0.001 for all). By contrast, a decrease was observed in the average length of stay and in the readmission rates, from 9.4 to 8.5 days and from 18.3% to 16.4%, respectively (*p* < 0.001 for both).

### 3.2. Hospital Admissions

Admissions caused by active SLE, CVD, infections, neoplasm or VTED are shown in Table 2. During the whole study period, active SLE was the main cause of admission (31.6% of cases), while infections accounted for 10.9%, CVD for 7.1%, neoplasms for 4.2%, and VTED for 1.1%. Admissions due to SLE decreased over the years, being the cause of 47.1% of the admissions in 1997–2000, 38.7% in 2001–2005, 29% in 2006–2010, and 20.8% in 2011–2015 (*p* < 0.0001 when comparing each period with the previous). On the contrary, hospitalizations due to CVD (from 5% in the first period to 8.6% in the last), infection (from 8.2% to 13%), and neoplasm (from 2.4% to 5.5%) increased over the time (*p* < 0.001 in all cases). Admissions due to VTED did not suffer significant variations. Overall, patients whose admissions were attributable to SLE were younger (mean 37.6 years) when compared to the other causes (*p* < 0.001 for all), and their mean age rose from the first to the fourth period in all the subgroups (*p* < 0.001).

### 3.3. Deaths

During the study period, 2786 patients died (6.41% of all the patients). Altogether, the mortality rate per admission was 2.79%. As shown in Table 1, the mortality rates, both overall and per admission, increased significantly after the year 2000.

SLE accounted for 13% of all deaths; CVD caused 18.5% of the deaths; infections 18.7%; neoplasm 11.7% and VTED 1.44% (Table 3). SLE decreased as a cause of death, from 24.2% in 1997–2000, 18.1% in 2001–2005, 12% in 2006–2010 to 6.4% in 2011–2015 (*p* < 0.001 when comparing each period with the previous). On the other hand, CVD (from 15.4% to 20.4%, *p* = 0.04), infection (from 14.3% to 21.1%, *p* = 0.005), and neoplasm (from 7.4% to 13.8%, *p* = 0.002) increased as causes of death from the first to the fourth period of study. Again, VTED remained unchanged throughout the study period.

Comparisons of the mortality rates, for each cause of admission, are shown in Table 3. The overall mortality rate was 1.2% for admissions attributable to SLE, significantly lower than in admissions due to CVD (7.3%), infections (4.8%), neoplasm (7.8%), and VTED (3.7%) (*p* < 0.001). There were no significant variations in the mortality rates when the different periods were compared. Patients who died because of SLE were significantly younger (mean age of 54.5 years) than those who died due to other causes (*p* < 0.001 for all). Indeed, the mean age of deceased patients significantly rose during the study period (from 58.5 years to 67.2, *p* < 0.001), reflecting the older age of patients dying from causes other than lupus activity (Table 3).

### 3.4. Subgroup Analysis

The subgroups comprised in the main categories were studied (Appendix A). Overall, coronary artery disease, cerebrovascular disease, and heart failure were the main determinants of CVD admissions and deaths. As shown in Figure 1A, all the CVD subgroups, except for arterial thromboembolism, increased as causes of admission overtime (*p* < 0.001 when the first and the fourth period were compared). On the other hand, no significant variations were found when the different cardiovascular causes of death were compared between different periods (Figure 1B). Respiratory infections were by far the main cause of admission and death (4.1% and 7% of all the admissions and deaths, respectively) in the infection group. All the infectious causes of admission increased when the first and the last period were compared (*p* < 0.02), except for tuberculosis, which actually decreased over time (from 1.5 % to 0.2%, *p* < 0.001) (Figure 2A). It is noteworthy that only sepsis significantly increased as a cause of death (from 0.3% to 1.2%, *p* < 0.001) (Figure 2B). The increasing admissions due to solid organ neoplasm (from 1.1% to 3.8%, *p* < 0.001) and cancer-related deaths (from 6% to 10.5%, *p* < 0.01) were the main determinants of the highest burden in morbidity and mortality of the neoplasms in the subsequent periods (Figure 3). Hematological neoplasms were, however, a less important cause of admissions and death.

## 4. Discussion

Robust evidence supports that mortality is two- to five-fold higher in SLE patients than in the general population [4,8,9,10,11]. In fact, SLE has been identified as the 10th highest cause of death among 15- to 24-year-old women [12]. However, mortality rates and specific causes of death have varied throughout the last decades [4,13,14,15,16]. Other groups have previously analyzed the causes of admission in SLE patients worldwide, with different results limited in many cases by sample size, monocentric design, and short follow-up periods [17,18,19,20,21,22,23,24,25]. Our study analyzes data from 99.859 admissions of SLE patients between 1997 and 2015, thus offering quite a wide view of the whole picture in Spain.

According to our analysis, hospital admissions and deaths due to active SLE decreased dramatically from the late 1990’s to the 2010–2015 period. Several studies have identified SLE itself as the main cause of hospital admission, with a wide range between 8.1 and 86.3% [17,18,19,20,21,22,23,24,25,26]. On the other hand, other studies have shown a significant downward trend in SLE-related admissions [19,27,28,29]. Recently, Anastisou et al. reported a decreasing risk of inpatient death in US lupus patients from 2006 to 2016 [29].

In our study, the reduction in SLE-related admissions and deaths was accompanied by a parallel increase in admissions and deaths due to CVD, infections, and neoplasm. In particular, CVD admissions, mainly determined by coronary and cerebrovascular disease and heart failure, rose from 5% to 8.6% of the total, and more than 20% of the deaths in the last period were of cardiovascular cause. Such an increase in the cardiovascular burden of SLE has been previously reported [12,15,16], and Piga et al. have also confirmed an increasing trend in admissions due to stroke and acute coronary syndrome between 2001 and 2012 [28].

According to our data, infections were the other main cause of death among Spanish SLE patients between 2011 and 2015. Previous studies have identified infections as an important cause of admission [19,23,25,30] and death [4,15,16,31,32,33,34]. Selvananda et al. reported that infections were concurrent with SLE flares in 41.1% of admissions, reflecting that both situations tend to coincide, and emphasizing the importance of the rational use of immunosuppressants [24]. Similarly to our findings, a recent study in the US showed that hospitalization rates due to infection significantly increased in the period 2015–2016 compared to 1998–2000, with sepsis overtaking pneumonia as the most common infection [35]. Both infections and CVD have been a major concern in longstanding SLE, with a significant impact on morbidity and mortality [1,2,36,37]. Infections are more common within the early phases of the disease, and are strongly determined by immunosuppressive treatment, whilst CVD tends to happen later and is mostly related to chronic inflammation, irreversible organ damage, and cumulative drug toxicity [3,23,38,39].

Despite the increased risk of malignancy in SLE patients, neoplasms have not yet been uniformly identified as a main cause of admission or death [40,41]. We found, in this study, that neoplasms increased as a cause of admission from 2.4% in 1997–2000 to 5.5% in 2010–2015 and became the third cause of death (from 7.4% to 13.8%). These changes were mainly due to solid organ tumors and might also be attributable to the ageing of SLE patients previously exposed to immunosuppression, a well-known risk factor for cancer development [42]. In addition, some of these drugs, such as azathioprine, cyclophosphamide, and mycophenolate, have been related to solid and hematological neoplasms themselves [43]. Similar evolution has been found from series published in the early 2000s when 2.3% of patients developed malignancy and 5.9% died because of cancer [3], and more recent studies, in which neoplasms caused over 13% of deaths in lupus patients [9,30], even being the first cause of death in some of them [44]. However, other authors have found neoplasms to be a stable, or even decreasing, cause of admission [19,28]. Obviously, these striking findings should be assessed deeply, and they merit further epidemiological studies to clarify the impact of neoplasm in SLE patients.

No significant variations in VTED-related admissions and deaths were found over the study period, similar to what has already been described [10,17].

Overall, the mortality rate in our population was 6.41%, with a 2.79% mortality rate per admission, which is in agreement with the literature [5,6,19,26,30]. However, it is noteworthy that in our registry, mortality rates rose after the year 2000, despite the improved survival observed in SLE patients over the last decades by other studies [6,13,14,15]. This apparent paradox could be explained by the increasing age of the admitted patients over the study periods, with the resultant decrease in SLE-related deaths relative to other causes. In other words, patients live longer due to better control of the disease, leading to late mortality related to CVD and malignancy.

Our study presents some limitations. Firstly, all the data come from a nationwide database, so the diagnoses could not be verified by the authors. Secondly, this analysis was performed considering hospital admissions, with a resultant limitation in power and potential selection bias. However, we mainly evaluated categorical variables, such as admission due to SLE, cardiovascular disease, infection, neoplasms, and deaths, which are difficult to misclassify. In parallel, SLE prevalence could not be properly assessed in the databases. Therefore, the rate of the events could not be elucidated and only admissions or deaths could have been compared. Secondly, we were unable to retrieve additional variables, such as CV risk factors, treatments, and the presence of antiphospholipid syndrome or specific organic SLE involvement. In this regard, data regarding hydroxychloroquine treatment, steroid-sparing immunotherapies, and prior steroid exposure would have yielded interesting data that could have supported our conclusions. Finally, we were only able to study the period between 1997 and 2015, due to the change in the registry diagnostic codification to ICD-10 after that year. On the other hand, our study offers a nationwide analysis with a large sample size and a long study period, with consistent results confirming the different trends shown in recent, smaller studies.

## 5. Conclusions

In conclusion, our large-scale study confirms the chronification and ageing of SLE patients. As a direct consequence, CVD, infections, and neoplasms have risen as admission causes, and have even surpassed lupus itself as the main cause of mortality in the last two decades. It is, thus, important to implement actions directed to mitigate the impact of such groups of diseases, including the extensive use of antimalarials and of lower doses of oral glucocorticoids, which can decrease infections and cardiovascular damage [45,46].

## Figures and Tables

**Figure 1 jcm-10-05749-f001:**
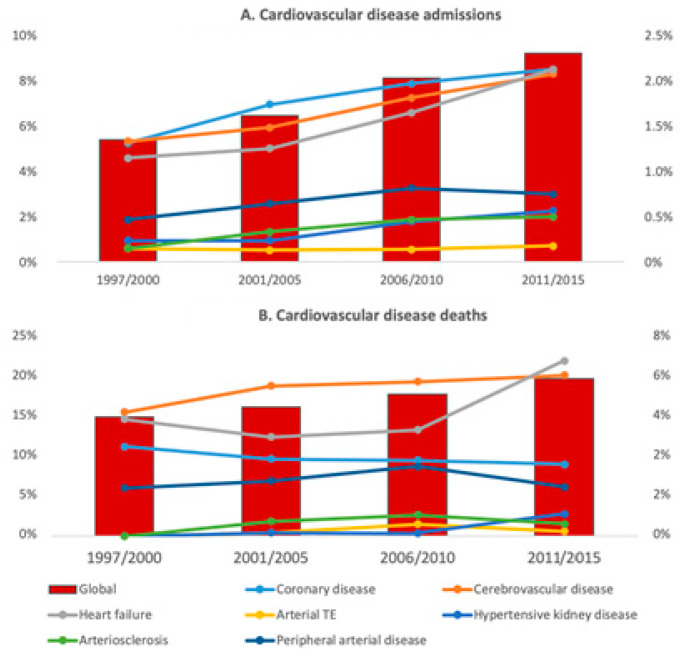
Cardiovascular disease admissions and deaths. The figure shows the trends in admissions (**A**) and deaths (**B**) for cardiovascular disease. Data are expressed as a percentage of total admissions and deaths for cardiovascular disease (left ordered axis) and for the subgroups that comprise cardiovascular disease (right ordered axis).

**Figure 2 jcm-10-05749-f002:**
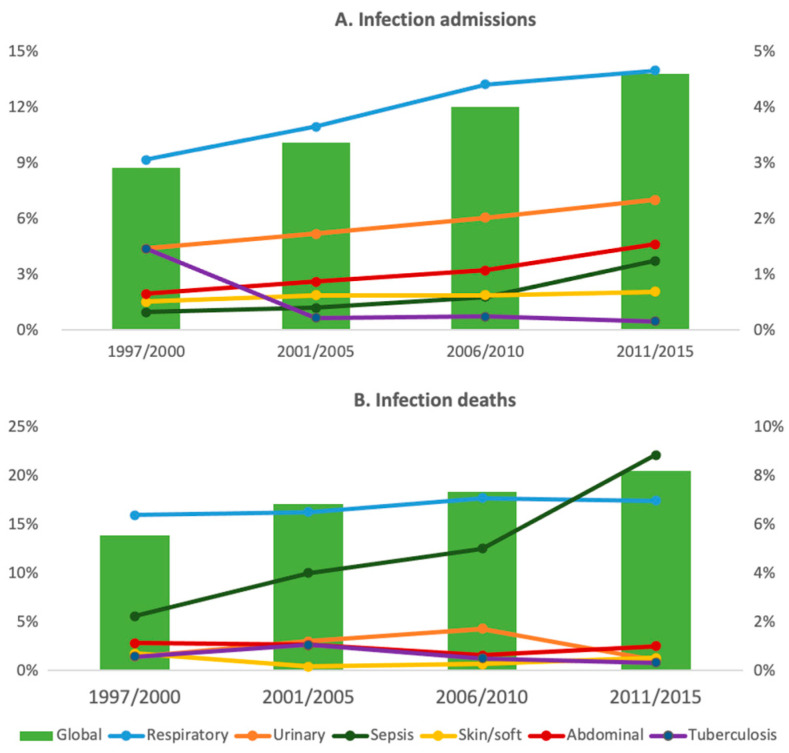
Infectious disease admissions and deaths. The figure shows the trends in admissions (**A**) and deaths (**B**) because of infection. Data are expressed as a percentage of total admissions and deaths for the overall infections (left ordered axis) and for the subgroups that comprise infectious disease (right ordered axis).

**Figure 3 jcm-10-05749-f003:**
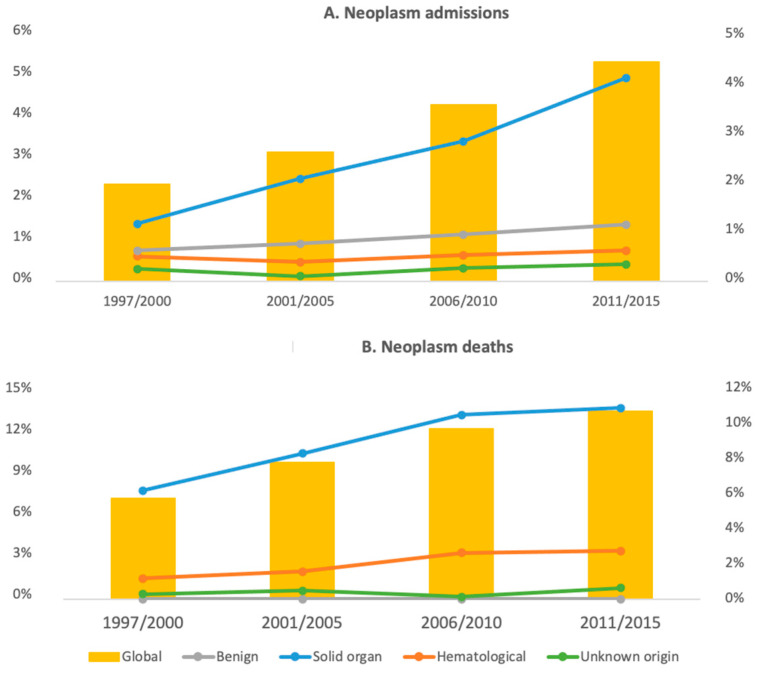
Neoplasm admissions and deaths. The figure shows the trends in admissions (**A**) and deaths (**B**) because of neoplasm. Data are expressed as a percentage of total admissions and deaths for the overall neoplasms (left ordered axis) and for the benign, solid organ, hematological and unknown origin neoplasm (ordered axis).

**Table 1 jcm-10-05749-t001:** Main demographic characteristics in SLE hospitalized patients.

	**Overall**	**1997–2000**	**2001–2005**	**2006–2010**	**2011–2015**
Patients (*n*)	43,432	8304	12,348	15,051	17,257
Gender female (%, CI)	83.3 (83.1–83.6)	82.8 (82.2–83.4)	83 (82.5–83.5)	83.4 (83–83.9)	83 (82.6–83.4)
Age (years) (Mean, SD)	46.5 (18.7)	41.1 (18.1)	44 (18.3) *	46.6 (18.5) *	51 (18.3) * T
Admissions (*n*)	99,859	15,807	24,204	27,781	31,977
Average stay (days) (Mean, SD)	9.1 (13.1)	9.4 (12.7)	9.5 (14.2)	9.3 (12.8)	8.5 (12.6) * T
Readmission rate (%, CI)	17.6 (17.4–17.8)	18.3 (17.7–18.9)	18.6 (18.1–19.1)	17.7 (17.3–18.2) *	16.4 (16–16.8) * T
Deaths (*n*, %)	2786 (6.41)	351 (4.22)	659 (5.34)	798 (5.30) *	978 (5.67) * T
Mortality rate per admission (%, CI)	2.79 (2.7–2.9)	2.22 (2.0–2.46)	2.72 (2.52–2.94) *	2.86 (2.67–3.07)	3.06 (2.87–3.25) T

SD: standard deviation, CI: confidence interval. *: *p* < 0.05 when compared with the previous period. T: *p* < 0.001 when compared with the first period.

**Table 2 jcm-10-05749-t002:** Rate of SLE hospitalized individuals by study period and clinical conditions.

	**Overall**	**1997–2000**	**2001–2005**	**2006–2010**	**2011–2015**
Active SLE					
Admissions *n* (%)	31,539 (31.6)	7440 (47.1)	9354 (38.7) *	8088 (29) *	6657 (20.8) * T
Age (years) (Mean, SD)	37.6 (16.5) **	35.9 (16.3)	37.1 (16.3)	37.9 (16.5)	40 (16.8) T
Cardiovascular disease					
Admissions *n* (%)	7065 (7.1)	789 (5)	1448 (6) *	2096 (7.5) *	2736 (8.6) * T)
Age (years) (Mean, SD)	59.5 (16.9)	55.5 (17.6)	57.1 (16.6)	59.6 (16.7)	61.6 (16.6) T
Infection					
Admissions *n* (%)	10,865 (10.9)	1292 (8.2)	2295 (9.5) *	3137 (11.3) *	4141 (13) * T
Age (years) (Mean, SD)	52.1 (19)	47.9 (18.8)	49.7 (18.9)	50.8 (19.1)	55.6 (18.4) T
Neoplasm					
Admissions *n* (%)	4182 (4.2)	386 (2.4)	785 (3.2) *	1238 (4.4) *	1773 (5.5) * T
Age (years) (Mean, SD)	56.5 (14.3)	54.8 (14.4)	55.8 (14.9)	55.3 (14.2)	57.9 (13.9) T
Venous thrombo-embolic disease					
Admissions *n* (%)	1069 (1.1)	174 (1.1)	277 (1.1)	268 (1)	350 (1.1)
Age (years) (Mean, SD)	50.1 (19.0)	45.1 (18)	47.9 (18.4)	51 (18.9)	53.6 (19.3) T

SLE: systemic lupus erythematosus. SD: standard deviation. *: *p* < 0.05 when compared with the previous period. T: *p* < 0.05 when compared with the first period. **: When SLE was compared to the other causes.

**Table 3 jcm-10-05749-t003:** Causes of mortality during admission and mean age of SLE patients.

	Overall	1997–2000	2001–2005	2006–2010	2011–2015
Active SLE					
Deaths (*n*, % of all deaths)	363 (13)	85 (24.2)	119 (18.1) *	96 (12) *	63 (6.4) * T
Mortality rate (%)	1.2	1.1	1.3	1.2	0.9
Age (years) (Mean, SD)	54.5 (19.7) **	53.5 (19.9)	52.3 (19.6)	55.7 (18.9)	58.5(20.5)
Cardiovascular disease					
Deaths (*n*, % of all deaths)	515 (18.5)	54 (15.4)	110 (16.7)	146 (18.3)	199 (20.4) T
Mortality rate (%)	7.3	6.8	7.6	7	7.3
Age (years) (Mean, SD)	67 (16.6)	60 (19.5)	63.5 (16.9)	66.9 (16)	71 (14.8) T
Infection					
Deaths (*n*, % of all deaths)	522 (18.7)	50 (14.3)	116 (17.6)	150 (18.8)	206 (21.1) T
Mortality rate (%)	4.8	3.9	5.1	4.8	5
Age (years) (Mean, SD)	64.5 (17.4)	62.5 (18.6)	62.5 (18.3)	61.9 (18.2)	68 (15.4) T
Neoplasm					
Deaths (*n*, % of all deaths)	327 (11.7)	26 (7.4)	66 (10)	100 (12.5)	135 (13.8) T
Mortality rate (%)	7.8	6.7	8.4	8.1	7.6
Age (years) (Mean, SD)	63.6 (13.6)	63 (11.7)	67.1 (12.6)	61.6 (14.6)	63.6 (13.4)
Venous thrombo-embolic disease					
Deaths (*n*, % of all deaths)	40 (1.4)	4 (1.1)	13 (2)	7 (0.9)	16 (1.6)
Mortality rate (%)	3.7	2.3	4.7	2.6	4.6
Age (years) (Mean, SD)	62.3 (16.1)	54.8 (5.8)	59.5 (17.8)	68 (16.5)	68.9 (15.3)

SLE: systemic lupus erythematosus. SD: standard deviation. *: *p* < 0.05 when compared with the previous period. T: *p* < 0.05 when compared with the first period. **: *p* < 0.01 when SLE was compared to the other causes.

## Data Availability

The data presented in this study are available on request from the corresponding author. The data are not publicly available since the database analysis has to be approved by the Spanish Ministry of Health.

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
