# Peer review of "Trends in Hospital Admissions and Death Causes in Patients with Systemic Lupus Erythematosus: Spanish National Registry"

_jcm, 2021, doi:10.3390/jcm10245749_

Round 1
Reviewer 1 Report
I would like to congratulate the authors on their exellent work.
I found only two small fault:
89th row: you wrote 83.1 % but 83.3 % is in table 1.
You write about the overall mortality in row 128-130, but these date are not included in table 3.
Author Response
Thank you very much for your comments.
- Indeed, 83.3% patients were female. This has been corrected.
- The mortality rate for each group and period has been added to the table.
Reviewer 2 Report
The authors described admission and death causes of SLE patients in large number of SLE patients from Spain (99,859,25 admissions from 43,432 patients with SLE). They have pointed out that cardiovascular disease (CVD), infections and neoplasm have become main causes of admissions and mortality. The information provided is informative and beneficial for readers. It will add more data from different region of the world and help the clinicians to follow-up their SLE patients.
Only minor points I would like to add.
- Could it possible to clarify the type of solid tumor among SLE individuals?
- table 2, the age shown is 379. I assume that it is a typo.
Author Response
Thank you very much for your comments.
1. We agree with the reviewer. While this information could possible to include, its extension and implications would probably excess the present manuscript length and objectives. In fact, we are performing a new and more profound analysis regarding neoplasms in SLE, since, as it was pointed out, cancer should be also considered a long-term complication of SLE.
2. The mean age for this period was 37.9 years-old. This has been corrected.